# Sustainable Waste Management Drilling Process in Fuzzy Environment

**Batyr Orazbayev [1], Saya Santeyeva [1], Ainur Zhumadillayeva [1], Kanagat Dyussekeyev [1], Ramesh K. Agarwal [2], Xiao-Guang Yue [3],* and Jiangchuan Fan [4,5]**

[1]  Faculty of Information Technologies, L.N.Gumilyov Eurasian National University, Nur-Sultan 010000, Kazakhstan; batyr_o@mail.ru (B.O.); saya_santeeva@mail.ru (S.S.); ay8222@mail.ru (A.Z.); abetovich@mail.ru (K.D.)
[2]  Washington University in St. Louis, St. Louis, MO 63101, USA; rka@wustl.edu
[3]  School of Sciences, European University Cyprus, Nicosia 1065, Cyprus
[4]  National Agricultural Information Technology Engineering Research Center, Beijing 100097, China; fanjc@nercita.org.cn
[5]  Beijing Agricultural Information Technology Research Center, Beijing 100097, China
*   Correspondence: x.yue@external.euc.ac.cy

**Abstract:** Sustainable management issues of waste during drilling oil wells in marine conditions, the process of disposal of drill cuttings in the conditions of deficiency, and fuzzy initial information using fuzzy inference system are investigated. Based on the conducted system analysis, the main criteria for controlling the process of re-injection of suspended drill cuttings were analyzed and selected. We described the technology of preparation and injection of drill cuttings slurry into the underground horizon. The method of modeling and management of the process of disposal of drilling cuttings in the marine environment in a fuzzy environment with the use of fuzzy inference system, which helps to overcome the problems of scarcity and fuzziness of the original information due to the knowledge and experience of experts are proposed. The scheme and structure of the elements of the fuzzy inference system based on the Mamdani algorithm are given. The implementation of the fuzzy output system procedure was carried out in MatLab using Fuzzy Logic Toolbox. For the purpose of sustainable waste management in the process of oil production of marine fields, waste management tasks are formulated as a fuzzy mathematical programming problem, which takes into account economic and environmental criteria and many production constraints that may be fuzzy. Since the vector of such criteria is characterized by inconsistency, the developed methods for solving the set tasks of sustainable management are based on various tradeoff schemes modified to work in a fuzzy environment. The novelty and originality of the developed methods lies in the fact that, unlike the well-known methods of similar methods for solving fuzzy problems, they are set and solved without conversion to a system of equivalent deterministic problems, with-out losing the main part of the collected fuzzy information. This allows, through the full use of the original fuzzy information, to obtain a more adequate solution to the fuzzy problem of the real problem under production conditions.

**Keywords:** drill cuttings; sustainable waste management; waste management; utilization; suspension injection system; fuzzy inference system; membership function; economic and environmental criteria; tasks of fuzzy mathematical programming

## 1. Introduction

Oil operations carried out during the commercial development of oil fields in the Kazakhstan sector of the Caspian Sea (KSCS): well drilling, oil and gas production, and transportation have a negative

effect on the sea's water basin, leading to an increase in waste. Therefore, the issues of effectively addressing the issues of oil operations sustainable waste management during the development of oil and gas fields KSCS is an important scientific and technical challenge [1,2]. To improve the state of the KSCS water basin and create favorable conditions for marine organisms, an effective waste management system is needed, which will be created on the basis of scientifically based methods. Since, when creating such systems, problems of shortage often arise, the fuzziness of the initial information [2] is especially important to develop and use a management system that is operable in a fuzzy environment [3]. Such systems are based on fuzzy information in the form of formalized with the methods of fuzzy set theory (FST), knowledge and experience of a person, expert decision maker (EDM) [3–7].

Sustainable waste management of oil operations, which are formed in the process of construction and installation works, at the stages of exploration and production drilling and during the operation of facilities of direct and auxiliary infrastructure, is an inevitable and integral part of the technological process at all its stages [8–10]. The main criteria for the management of waste oil operations can include: efficiency of recycling processes; ensure environmental safety and economic efficiency on the protection of the aquatic environment [11]. At the same time, it is necessary to comply with the current and relevant legislation of the Republic of Kazakhstan and the degree of prescriptions. As well designs become more complex, it is necessary to make greater efforts to meet stringent standards for waste disposal while meeting the requirements for drilling efficiency. Modern advances in drilling fluids, chemical injections, and drilling waste disposal technologies allow the use of the most effective compositions of drilling fluids, chemical injections, and simultaneously with efficient removal of drill cuttings [11]. Unlike famous works [11,12], in this paper, standards and limitations when setting optimization tasks and sustainable waste management can be seen as limitations that can be fuzzy and limit the choice of technology.

In well-known formulations and solution methods fuzzy problems at the formulation stage are replaced by a system of equivalent clear problems based on a set of level $\alpha$ [13]. This approach allows us to get an approximate solution to the fuzzy problem, but a significant part of the original collected fuzzy information is not used and is lost, which will lead to a decrease in the adequacy of the obtained solutions. The advantage of the fuzzy approach proposed in this paper to the formulation and solution of fuzzy mathematical programming problems is that the problem is posed and solved in a fuzzy environment without first converting to equivalent clear problems. This allows us to save and maximize the use of the original fuzzy information. And this, in turn, allows us to obtain an adequate solution to the problem of managing production waste in a fuzzy environment, that is, in real situations.

The method of re-injection of drill cuttings into the reservoir is an important method of waste disposal during exploration and production of oil and gas. Injecting drilling cuttings and other drilling waste into the deep plates is often the most cost-effective way to dispose of oil operations waste and proven environmental safety, especially in the sea conditions, in more ecological sensitive areas such as the Kazakh sector of North Caspian [13].

The ban on the dumping of waste into the environment in the ecologically sensitive region of the Northern Caspian does not allow for the usual burial of drill cuttings or solid waste on the seabed. In the case of the inadmissibility of the dumping of waste into the environment during oil operations there are two possibilities [14]:

(1) Waste can be collected and redirected to the seaside (skip-and-ship process) and further buried at a special polygon. By its nature, this method may present security risks, owing to the recurring for loading and unloading solid and liquid waste. For this method it is also necessary to have ground polygons of considerable capacity. In fact, this method represents only the transfer of waste from one natural environment (marine) to another (land);

(2) Re-injection of waste into the reservoir, i.e., returning them to where they were taken from. The preference of this method, i.e., re-injection of drill cuttings due to the fact that this method:

- Satisfies the principle of non-dumping of waste into the environment due to the exclusion of dumping at sea;
- Eliminates the need for loading and unloading both at sea and ashore, as well as the need for polygon capacity;
- Returns drilling waste to their natural location;
- Is an ecologically friendly method of waste management in remote locations in cases where land disposal is not possible;
- Eliminates or reduces the risks associated with safety and transportation during repetitive loading and unloading of waste during collection and transfer to the shore.

In this paper, we propose an approach to the management of drill cuttings based on the method of re-injection of waste into the reservoir. In this regard, scientifically substantiated methods for implementing the selected method of waste disposal have been investigated. Since this raises the problem of the fuzziness of some part of the initial information, a fuzzy approach is reasonably applied, which allows taking into account the knowledge, experience and intuitions of subject matter experts.

If a suitable reservoir exists, carrying out the re-injection of drill cuttings requires careful assessment, development of all details of operations, execution, monitoring, and control of the injection process to reduce possible risks, localize the injected material within specified limits and improve the efficiency of the waste disposal process. Monitoring the injection process is also appropriately needed to demonstrate the reliable conservation of the material and determine the technical indicators on the basis of which the development and implementation of this process could be updated and improved.

However, it should be noted that, when drilling under certain conditions, there may be no adjacent formations suitable for injection and reliable preservation of material within the formation. It is clear that in such cases, the method of drilling sludge re-injection is not suitable and will have to use other methods of waste disposal [15].

The aim of the study is to analyze the problems of sustainable management of drill cuttings during drilling in marine conditions based on literature data and to develop an approach to their solution using a fuzzy approach. The contribution of the expected results of the study is to solve the problems of waste management in marine conditions (for example, marine fields of the Caspian Sea) and to increase the adequacy of the sustainable management process by using the experience and knowledge of expert experts (fuzzy information) and the apparatus of the theory of fuzzy sets.

## 2. Method of Re-Injection of Drill Cuttings and Its Technology

When using the method of re-injection for the purpose of disposal of drill cuttings, a suspension of a certain density, obtained as a result of additional grinding of drill cuttings and the addition of drainage, oil-containing, and, if necessary, sea water, is pumped into the underground horizons.

The well-known standard re-injection technology involves mixing the sludge with sea water followed by grinding or other mechanical treatment until a stable viscous suspension is injected into the reservoir under pressure through a specially drilled for this well or through the existing exploitation well in annular space. As a result, hydraulic fracturing occurs with the effective burial of the suspension in the cracks. As a rule, at the end of re-injection in the well or in the annular space, a cement bridge is installed.

Thus, when disposing of drill cuttings and oily waste, the re-injection sludge has a minimal impact on the environment while simultaneous provision economic efficiency. In addition, this method of utilization of drill cuttings, drainage, and oil-containing waters is more rational in connection with the exception of long and expensive operations associated with the transportation of drill cuttings to the shore, their processing at thermal desorption facilities, and burial at landfills. At the same time, the injection of suspended drill cuttings into the underground horizons in the sea increases the risk of environmental pollution in emergency situations during these operations. In order to eliminate emergencies and risks, a system for managing the process of injection of drill cuttings is necessary.

Objects of ground equipment of the drilling sludge injection system of the considered technology and the option of utilization of drilling sludge can be represented as a Figure 1.

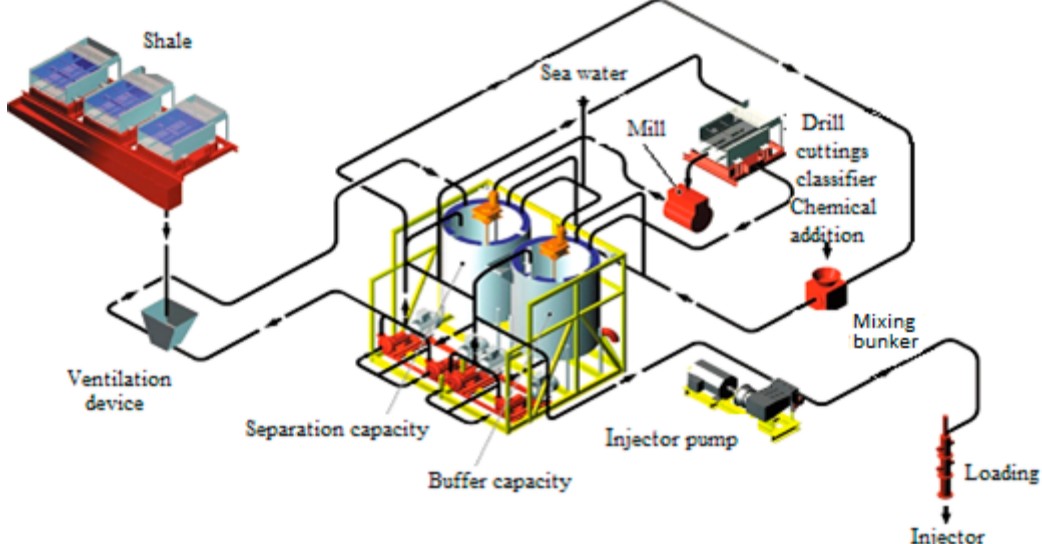

**Figure 1.** Objects of ground equipment of drilling cuttings injection system.

As can be seen from the figure, the installation for the preparation and injection of a drill cuttings suspension in the underground horizon included [16,17]:

- Shale shakers for preliminary classification of drill cuttings according to their grain size;
- Milling mill for grinding a large fraction;
- Separation capacity with mixing device;
- Bunker and high-speed mixer of the input of shale to the suspension of cuttings;
- Loading device and mixer for the input of chemicals into the suspension of drill cuttings;
- Pumping unit for circulation of suspension drilling cuttings;
- Buffer capacity of drill cuttings slurry prepared for injection;
- Wellhead device with an injector for pumping slurry into underground horizons.

The scheme of injection of suspended drilling sludge into the underground horizons into the arrays of low-permeable rocks with the formation of a system of artificial cracks, which are filled with drilling waste during the injection process, is shown in Figure 2. Methods of waste management predetermine the presence of objects at landfills necessary for the tasks of environmentally safe waste disposal.

Thus, waste disposal sites should have:

1. Maps for the disposal of waste, including receiving and disposing of liquid effluent,
2. Storing solid waste, burying;
3. Unit for thermal desorption of drill cuttings;
4. Installation for the incineration of solid waste, oil sludge, and excess sludge from wastewater treatment plants;
5. Domestic water treatment plant;
6. Installation of purification of oily water;
7. Open storage shed for waste in containers;
8. Septic tanks for liquid waste.

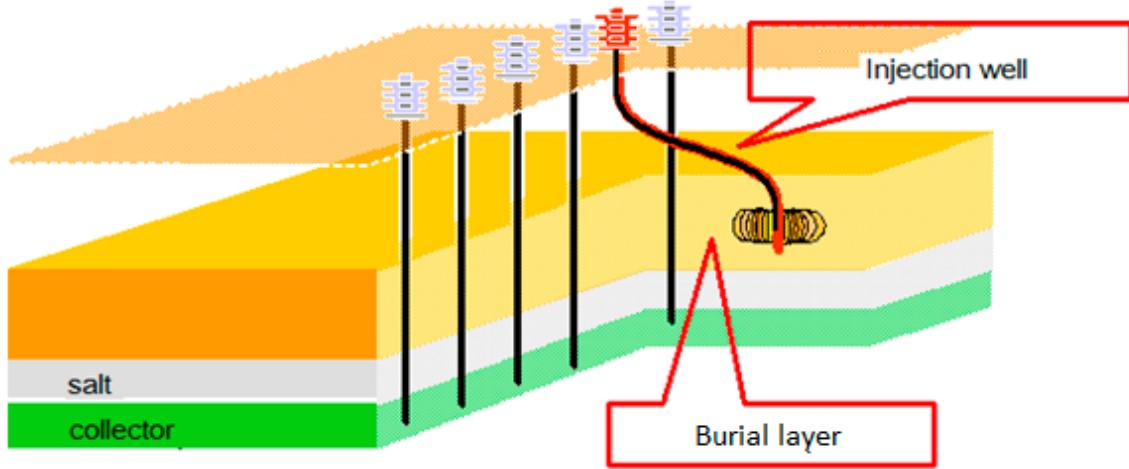

**Figure 2.** Scheme of injection of drill cuttings in the underground horizon.

The drilling waste management options used in world practice can be divided into 3 categories: discharge into the sea, injection into absorbing horizons at the work site, and removal to land with subsequent processing and disposal. In the conditions of the Kazakhstan sector of the Caspian Sea, the second option is recommended as the most acceptable option, i.e., injection into absorbing horizons at the work site.

To study the method of reverse injection of drill cuttings and its technology, it is required to develop a system of mathematical models that describe the processes of injection of drill cuttings and control these processes. As shown by the practice of collecting and processing the necessary information to develop mathematical models of drill cuttings management processes in marine oil and gas fields of KSCS, there is a shortage of reliable theoretical and statistical information needed to build analytical or statistical models. The organization and conduct of theoretical and experimental studies of these processes in order to collect the missing part of the information is not economically feasible. But, during the analysis, it turned out that there are experienced specialists in the subject area, i.e., drillers, ecologists, and engineers who can adequately describe in a natural language the processes of pumping drill cuttings and controlling these processes. They can verbally describe what are the main parameters and how they affect the process of disposal of drill cuttings. Based on such fuzzy information using the theory of fuzzy sets, one can construct rule bases, i.e., linguistic models of the drilling waste disposal process, which are relatively easy to implement using the MatLab Fuzzy Logic Toolbox application.

## 3. Fuzzy Output System to Sustainable Manage the Process of Disposal of Drill Cuttings

Consider the sustainable management of the process of disposal of drill cuttings based on the modeling method and fuzzy output systems.

When modeling and managing the process of disposal of drill cuttings, problems of short-age of initial quantitative information, and uncertainty due to the fuzzy information available often arise. In this case, a fuzzy description of the process of disposal of drill cuttings can be obtained on the basis of human knowledge and experience (subject-matter experts) using expert assessment methods and theories of fuzzy sets [3,4,18–22].

This paper proposes a modeling method and managing the process of disposal of drill cuttings in marine environments in a fuzzy environment using a fuzzy inference system that conducive to overcoming problems of shortage and fuzziness of initial information due to the knowledge and experience of specialist-experts. Diagram of modeling and management of the process of disposal of drill cuttings in the fuzzy output system is shown in Figure 3.

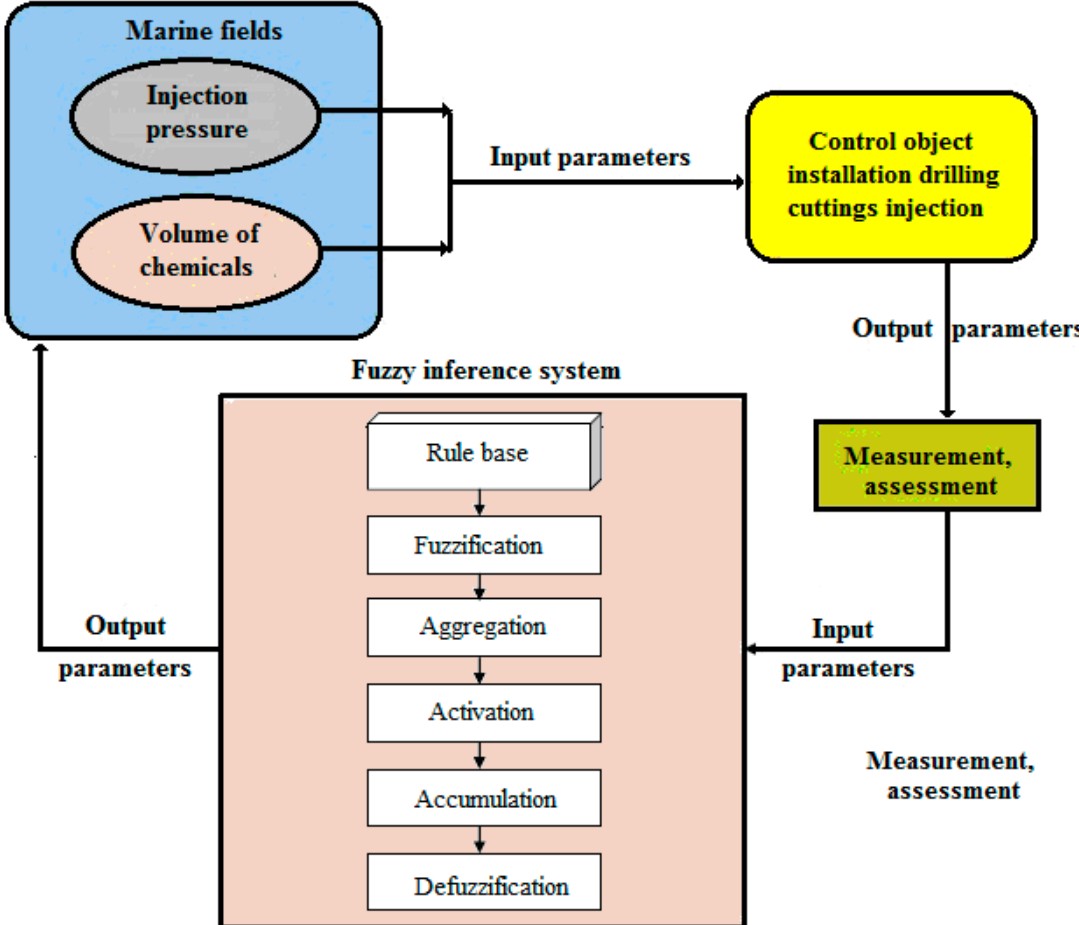

**Figure 3.** The structure of the elements of the fuzzy inference system.

As can be seen from Figure 3, the input parameters for the control object, i.e., for the installation of injection of drill cuttings in a marine field are the values of the injection pressure and chemical reagents volume and the output parameter of the installation is an indicator of the utilization of drill cuttings. Since these parameters are difficult to quantify quantitatively, it is advisable to evaluate their values using expert evaluation (in the measurement, evaluation block). Then, the fuzzy output system receives fuzzy values of the estimated input parameters, and the fuzzy value of the percentage of utilization is obtained from the output of the system. Depending on the output value of the fuzzy inference system and the rule base control actions are generated.

As known, there are several fuzzy inference algorithms in the production rules system. Among them, the most widely used algorithms of Larsen, Mamdani, Tsukamoto [23–25]. To solve our tasks waste management problem we used the idea of the most universal fuzzy inference algorithm proposed by E. Mamdani (Ebrahim Mamdani) [24].

The contribution of this part of the article is to apply a fuzzy approach to solving the problems of managing drill cuttings in a particular marine field of the Northern Caspian (Kashagan field) based on the adaptation of the Mamdani algorithm.

The Mamdani Algorithm consists of the following basic steps:

1) Formation of the rule base of fuzzy inference systems. The rule base is a set of fuzzy product rules, in which conditions and conclusions are formulated in terms of fuzzy statements. In our task, the input parameters (variables), the values of which are set outside the model of the fuzzy inference system, are: $\widetilde{a}_1$—"volume of chemical reagents" and $\widetilde{a}_2$—"injection pressure".

The output parameter (variable), the value of which is formed inside the model: $\widetilde{b}$ *plant performance—disposal of drill cuttings*.

For an abbreviated notation of the rules, we use the notation presented in Table 1. The universal sets (universes) of the given fuzzy parameters are given in Table 2.

**Table 1.** Description of the simulated parameters for the formation of rules.

| Description of the Level Values of the Fuzzy Parameters | Notation |
|---|---|
| High | *HG* |
| High of medium | *HM* |
| Medium | *MD* |
| Low of medium | *LM* |
| Low | *LW* |

**Table 2.** Universums for fuzzy parameters $\widetilde{a}_1$, $\widetilde{a}_2$, and $\widetilde{b}$.

| Fuzzy Parameter | Level Values of the Fuzzy Parameters | | | | |
|---|---|---|---|---|---|
| | *HG* | *HM* | *MD* | *LM* | *LW* |
| $\widetilde{a}_1$—volume of chemicals, liters | 40–50 | 30–40 | 20–30 | 10–20 | 1–10 |
| $\widetilde{a}_2$—injection pressure Pound/inch$^2$ | 525–575/ 1075–1125 | 475–525/ 1025–1075 | 425–475/ 975–1025 | 375–425/ 925–975 | 325–375/ 875–925 |
| $\widetilde{b}$—disposal of drill cuttings, in % | 85–100 | 60–85 | 40–60 | 15–40 | 1–15 |

Tables 1 and 2 show the 5 levels of values of fuzzy parameters (HG-High; HM-High of medium; MD-Medium: LM-Low of medium: LW-Low), i.e., values of linguistic variables, and Figure 4 shows 3 linguistic parameters $\widetilde{a}_1$ (volume of chemical reagents), $\widetilde{a}_2$ (injection pressure), and $\widetilde{b}$ (disposal of drill cuttings).

Table 2 shows universes for fuzzy variables, and the parameters of the membership function, i.e., the main points describing membership functions are obtained based on expert judgment. Then, smoothly connecting these points, disjoint graphs of the membership function are obtained, which are identical to the graphs of the membership function shown in Figure 5a–c.

The experts evaluated the main control points of the membership function, specifically: the extreme points where the membership functions take minimum values, the point in the middle of the universe where the membership functions take maximum values, and also the points between the maximum and minimum values of the membership function. Then, by smoothly connecting these points, graphic forms of triangular form membership function are obtained, which are easily approximated to analytical expressions. Moreover, in the MatLab Fuzzy Logic Toolbox application, such forms of membership function are available and it is convenient to apply them. In our case, these forms of membership functions, and the results of fuzzy modeling based on the constructed rule base, gave fairly adequate (similar) data with real data. This allows using the Fuzzy Logic Toolbox application to carry out fuzzy simulations with other fuzzy values of the input linguistic variables. Therefore, in this paper, a fuzzy approach is used.

Developed fuzzy output rules for a fuzzy inference system, i.e., linguistic models describing the relationship between input and output linguistic variables are presented in the form of the following fuzzy product rules:

**Rule 1.** *If "$\widetilde{a}_1$ is HG" and "$\widetilde{a}_2$ is HG", then "$\widetilde{b}$ is HM" $F_1$;*

**Rule 2.** *If "$\widetilde{a}_1$ is HG" and "$\widetilde{a}_2$ is HM", then "$\widetilde{b}$ is MD" $F_2$;*

**Rule 3.** *If "$\widetilde{a}_1$ is HG" and "$\widetilde{a}_2$ is MD", then "$\widetilde{b}$ is LM" $F_3$;*

**Rule 4.** *If "$\widetilde{a}_1$ is HG" and "$\widetilde{a}_2$ is LM", then "$\widetilde{b}$ is LW" $F_4$;*

**Rule 5.** *If "$\widetilde{a}_1$ is HG" and "$\widetilde{a}_2$ is LW", then "$\widetilde{b}$ is LW" $F_5$;*

**Rule 6.** *If "$\widetilde{a}_1$ is HM" and "$\widetilde{a}_2$ is HG", then "$\widetilde{b}$ is HM" $F_6$;*

**Rule 7.** *If "$\widetilde{a}_1$ is HM" and "$\widetilde{a}_2$ is HM", then "$\widetilde{b}$ is MD" $F_7$;*

**Rule 8.** *If "$\widetilde{a}_1$ is HM" and "$\widetilde{a}_2$ is MD", then "$\widetilde{b}$ is LM" $F_8$;*

**Rule 9.** *If "$\widetilde{a}_1$ is HM" and "$\widetilde{a}_2$ is LM", then "$\widetilde{b}$ is LW" $F_9$;*

**Rule 10.** *If "$\widetilde{a}_1$ is HM" and "$\widetilde{a}_2$ is LW", then "$\widetilde{b}$ is LW" $F_{10}$;*

**Rule 11.** *If "$\widetilde{a}_1$ is MD" and "$\widetilde{a}_2$ is HG", then "$\widetilde{b}$ is HM" $F_{11}$;*

**Rule 12.** *If "$\widetilde{a}_1$ is MD" and "$\widetilde{a}_2$ is HM", then "$\widetilde{b}$ is HM" $F_{12}$;*

**Rule 13.** *If "$\widetilde{a}_1$ is MD" and "$\widetilde{a}_2$ is MD", then "$\widetilde{b}$ is MD" $F_{13}$;*

**Rule 14.** *If "$\widetilde{a}_1$ is MD" and "$\widetilde{a}_2$ is LM", then "$\widetilde{b}$ is LM" $F_{14}$;*

**Rule 15.** *If "$\widetilde{a}_1$ is MD" and "$\widetilde{a}_2$ is LW", then "$\widetilde{b}$ is LW" $F_{15}$;*

**Rule 16.** *If "$\widetilde{a}_1$ is LM" and "$\widetilde{a}_2$ is HG", then "$\widetilde{b}$ is HM" $F_{16}$;*

**Rule 17.** *If "$\widetilde{a}_1$ is LM" and "$\widetilde{a}_2$ is HM", then "$\widetilde{b}$ is MD" $F_{17}$;*

**Rule 18.** *If "$\widetilde{a}_1$ is LM" and "$\widetilde{a}_2$ is MD", then "$\widetilde{b}$ is MD" $F_{18}$;*

**Rule 19.** *If "$\widetilde{a}_1$ is LM" and "$\widetilde{a}_2$ is LM", then "$\widetilde{b}$ is LM" $F_{19}$;*

**Rule 20.** *If "$\widetilde{a}_1$ is LM" and "$\widetilde{a}_2$ is LW", then "$\widetilde{b}$ is LW" $F_{20}$;*

**Rule 21.** *If "$\widetilde{a}_1$ is LW" and "$\widetilde{a}_2$ is HG", then "$\widetilde{b}$ is HM" $F_{21}$;*

**Rule 22.** *If "$\widetilde{a}_1$ is LW" and "$\widetilde{a}_2$ is HM", then "$\widetilde{b}$ is MD" $F_{22}$;*

**Rule 23.** *If "$\widetilde{a}_1$ is LW" and "$\widetilde{a}_2$ is MD", then "$\widetilde{b}$ is LM" $F_{23}$;*

**Rule 24.** *If "$\widetilde{a}_1$ is LW" and "$\widetilde{a}_2$ is LM", then "$\widetilde{b}$ is LM" $F_{24}$;*

**Rule 25.** *If "$\widetilde{a}_1$ is LW" and "$\widetilde{a}_2$ is LW", then "$\widetilde{b}$ is LW" $F_{25}$;*

here $F_1$, $F_2$..., $F_{25}$ are weights coefficients the degree of confidence in the validity of the subclauses. These coefficients take values in the range from zero to one.

2) The fuzzification of exogenous variables. Fuzzification is a procedure for finding the values of membership functions of fuzzy sets based on the initial data.

At this stage, the system receives many input variables $A = \{a_1, a_2 \ldots, a_{50}\}$ with known specific values, as well as the rule base formed at the previous stage of the algorithm.

Then, for each of the sub-conditions, there is a value from the equation:

$$b_i = \mu(a_i),\ i = 1, 2, \ldots, 50$$

The number of sub-conditions in the base of rules, in our problem is 50. Many exogenous values should be obtained in an external source an attitude to the fuzzy inference system. Fuzzification procedures and other algorithm procedures are implemented in MatLab using Fuzzy Logic Toolbox.

Figure 4 shows the editor window with the parameters of the fuzzy inference system.

Figure 5 shows the results of fuzzification fuzzy parameters: input parameters.

3) Aggregation of sub-conditions in fuzzy product rules. At this stage, the degree of truth of the conditions of each of the rules of the fuzzy inference system is determined. To find the degree of truth of the conditions of each of the rules of fuzzy products, you can use paired fuzzy logic operations. For example, in our case, for the sub-conditions related to each other by the operation "and", we find the minimal values of truth from all sub-conditions: $s_j = \min\{b_{l'}\}$, $j = 1, \ldots, k$ where $k$—number of rules in the system ($k = 25$), $i' \in [0; 1]$—numbers from the set of sub-conditions in which the j-th variable participates. In further calculations, as already, only those conditions are involved (active), the degree of truth of which is non-zero.

The upper part of the (first) panel shows the membership functions of fuzzy input and output variables, and the second panel (lower part) lists the parameters, universes, and other parameters of these fuzzy variables.

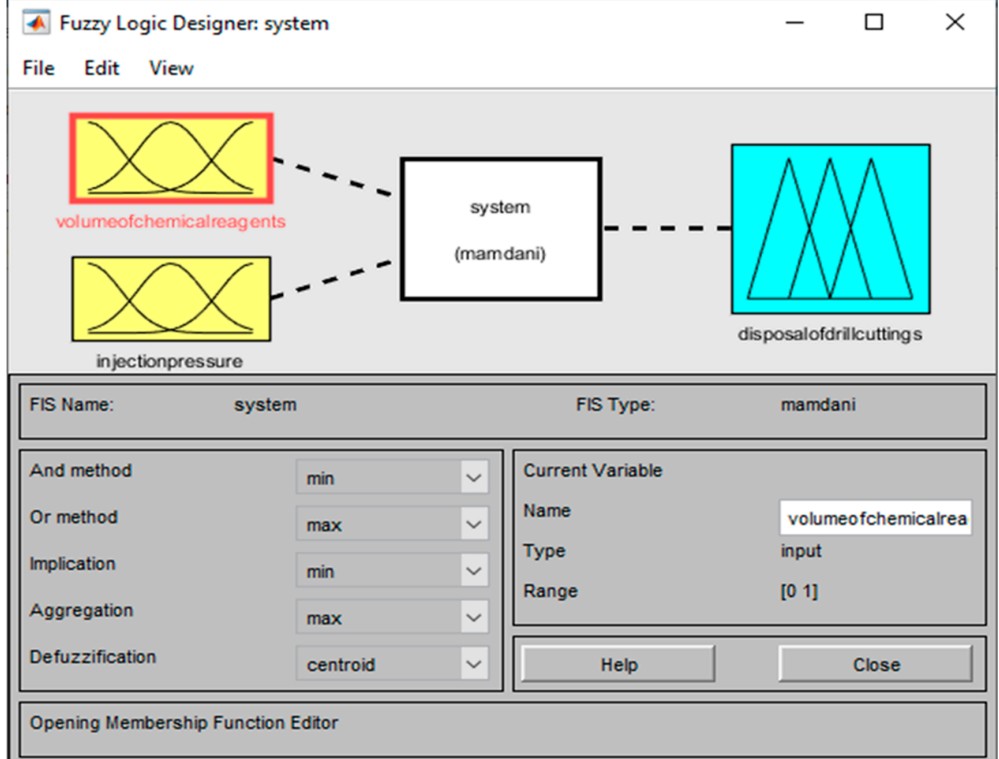

**Figure 4.** FIS*-Editor editor window for the problem to be solved. (* The editor of applications of the Fuzzy Logic Toolbox of the MatLab system.)

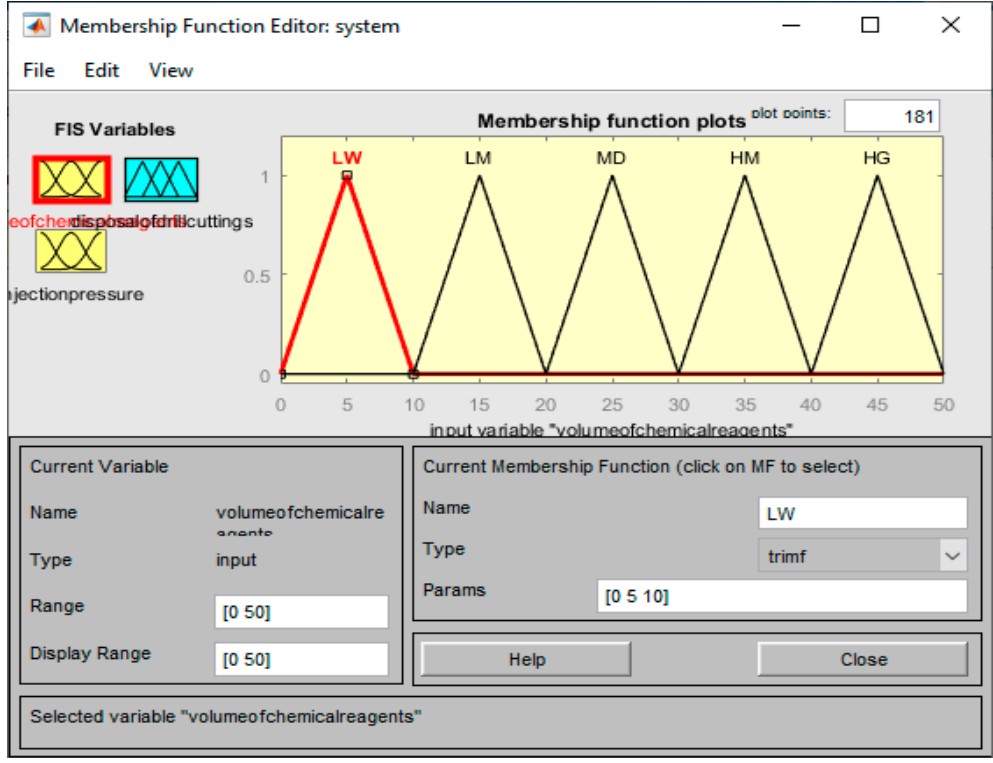

(**a**) Chemical reagents.

**Figure 5.** *Cont.*

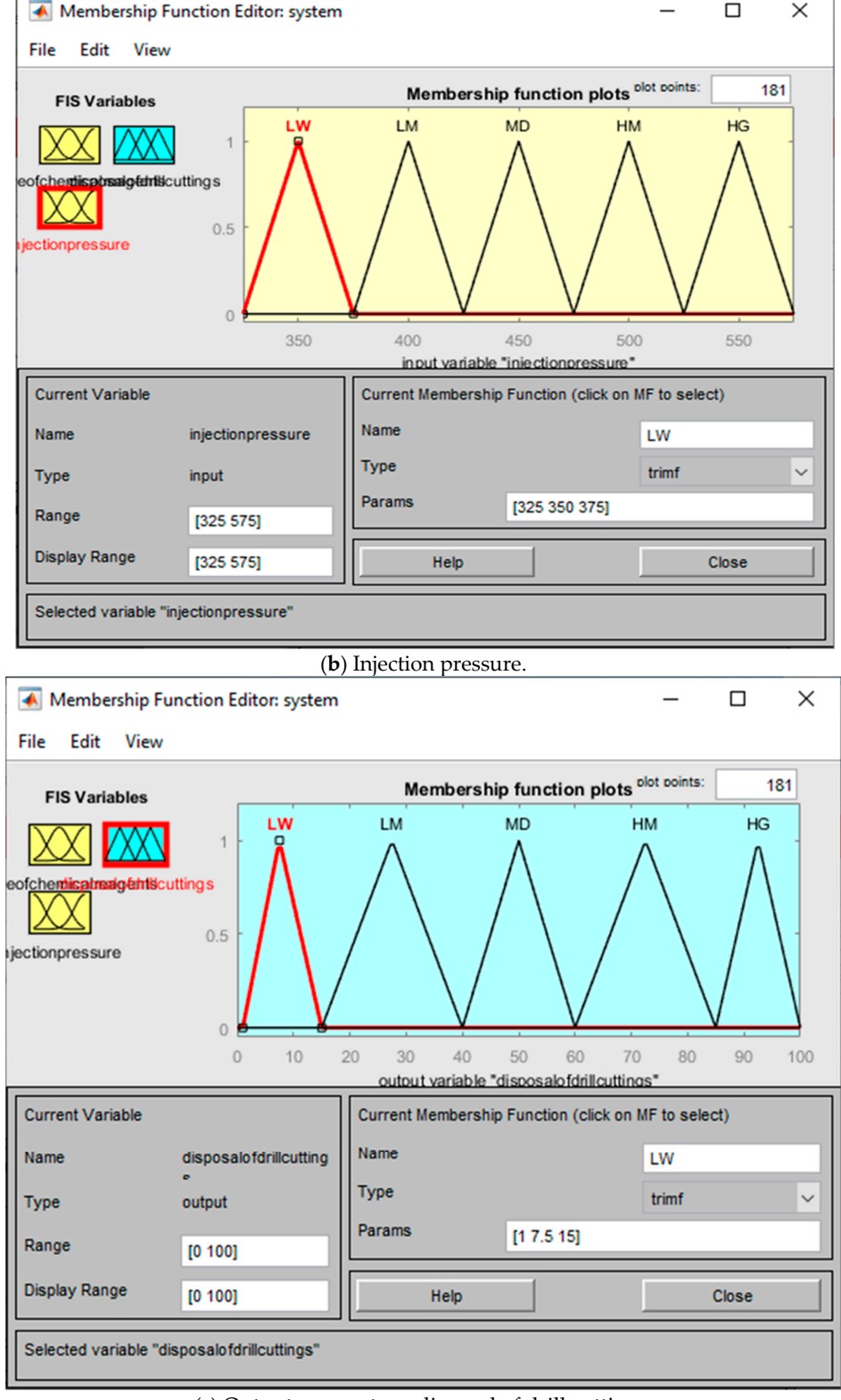

(**b**) Injection pressure.

(**c**) Output parameter—disposal of drill cuttings.

**Figure 5.** Membership functions of fuzzy parameters.

4) Activation of connections in fuzzy rules of products. At this stage, the degree of truth is determined for each of the sub-conclusions of the fuzzy inference system, and a fuzzy set is formed for each of the endogenous variables. If the conclusion consists of one sub-conclusion, then the degree of its

truth is determined by the algebraic product of the value $s_{j'}$ and the weight coefficient $F_{j'}$, $j' = 1, \ldots, l$ where l—the total number of sub-conclusions in the rule base.

After finding the set $D = \{d_1, \ldots, d_l\} = \{s_1 F_1, \ldots, s_l F_l\}$ the membership functions of each of the sub-conclusion for endogenous variables are determined by the formula [26]: $\widetilde{\mu}(w) = \min\{d_{j'}, \mu(w)\}$. Here $\widetilde{\mu}(w)$—the term membership function, which is the value of a certain endogenous variable defined on the universe $W$. As a result, each endogenous variable included in the individual sub-conclusion of the rules determines the membership functions of the fuzzy set of its values—$D^1, \ldots, D^l$.

5) Accumulation of conclusions. Accumulation of conclusions in fuzzy rules of products can be carried out according by the formula for combining fuzzy sets corresponding to the terms of the sub-conclusion related to the same output linguistic variables. The goal of this stage is to unite all the degrees of truth of the sub-conclusion to find the membership function of each of the endogenous linguistic variables of the set $W = \{w_1, \ldots, w_k\}$. The need for accumulation arises due to the fact that sub-conclusion that belong to one endogenous variable may belong to different rules. So, each of the endogenous variables $w_{i'} \in W$ and the fuzzy sets $D_{i'}^1, \ldots, D_{i'}^l$ relating to it are consistently investigated. The result of the accumulation of an endogenous variable $w_{i'}$ represents the union of fuzzy sets $D_{i'}^1, \ldots, D_{i'}^l$. As a result, for each endogenous linguistic variable, the final membership functions of fuzzy sets of their values, namely the totality of fuzzy sets $D_{i'}^1, \ldots, D_{i'}^l$, must be defined.

6) Defuzzification of endogenous (output) variables. The purpose of the final stage of the algorithm is to obtain a quantitative value for each of the endogenous linguistic variables of the set $W = \{w_1, \ldots, w_k\}$. So, each endogenous variable $w_{i'} \in W$ and its fuzzy set $\widetilde{D}^{i'}$ are consistently considered. As a result of defuzzification, the endogenous linguistic variable is defined as a normal quantitative value $y_{i'}$. To establish the value $y_{i'}$ used the defuzzification method—the center of gravity, to which the formula is used: $y = \int\limits_{Min}^{Max} w \cdot \mu(w) dw / \int\limits_{Min}^{Max} \mu(w) dw$, where y—the result of defuzzification; w—the variable corresponding to the endogenous (output) linguistic variable; $\mu(w)$—membership function of a fuzzy set corresponding to the output variable $w$ after the accumulation stage;

*Min* and *Max*—left and right points of the interval of the carrier of a fuzzy set of the considered output variable $w$. With the center of gravity defuzzification, the usual (not fuzzy) value of the output variable is equal to the abscissa of the center of gravity of the area bounded by the graph of the curve of the membership function of the corresponding output variable [27].

For single-point sets, the center of gravity can be defined as: $y = \sum\limits_{i=1}^{n} w_i \cdot \mu(w_i) / \sum\limits_{i=1}^{n} \mu(w_i)$. Here $n$—the number of one-point (one-element) fuzzy sets, each of which characterizes the only value of the output linguistic variable considered. Figure 6 shows a visualization window for a fuzzy inference.

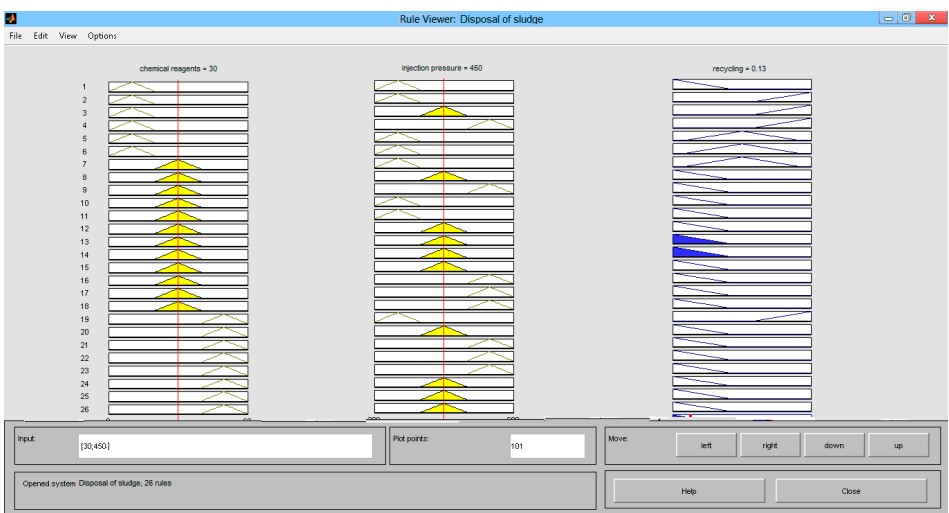

**Figure 6.** Fuzzy logical inference visualization in RuleViewer.

The input field shows the values of the input variables for which the logic output is performed. The "inputs-output" surface corresponding to the synthesized fuzzy system is shown in Figure 7.

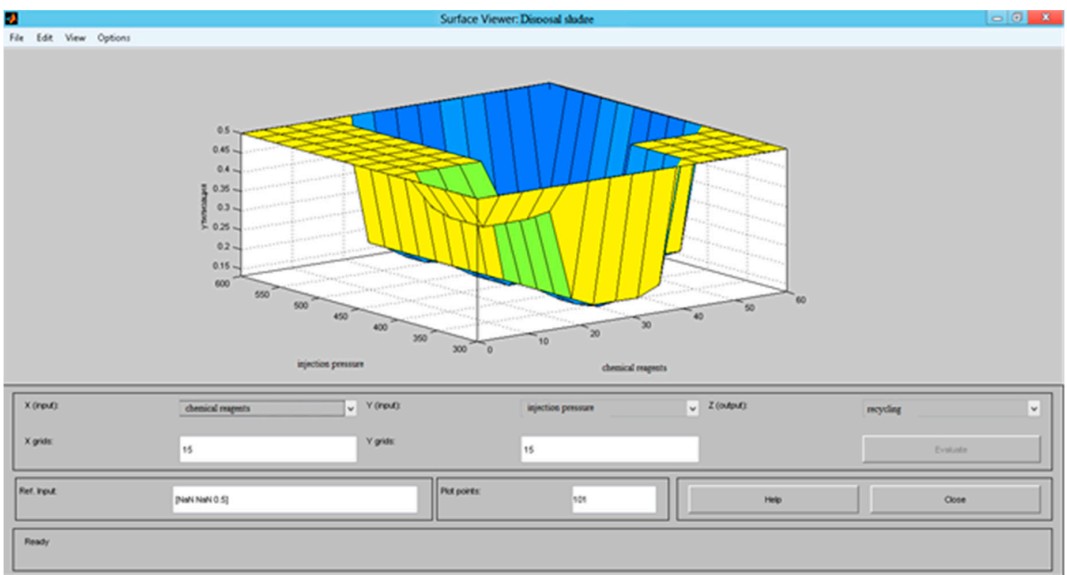

**Figure 7.** "Inputs-output" surface in the window Surface Viewer.

## 4. Tasks of Fuzzy Programming for Sustainable Waste Management in a Fuzzy Environment and the Development of Methods for Their Solution

### 4.1. Discussion of the Results

We formulate the decision-making tasks for the sustainable management of drilling waste in the form of fuzzy mathematical programming problems and propose heuristic methods for solving them, based on the use of mathematical models of the process and experience, knowledge, and judgments of a person—decision maker [28].

In the presence of uncertainty for a number of objective reasons, the use of probabilistic methods for solving decision-making and control problems is not justified. Moreover, in cases where there is reason to believe that the objects under study behave according to probabilistic laws, the lack of information, the inadequacy of statistical data, the inexpediency of their collection pushes other ways of describing real processes, to build non-statistical, for example, fuzzy approaches. The combination of various sources of fuzziness (in the criteria, restrictions, requirements for them) leads to various fuzzy problems of decision-making and control. Let us give some statements of the problems of sustainable control drill cuttings in a fuzzy environment and approaches to their solution [29–34]. With a fuzzy source information, the problem to be solved is formulated as a mathematical programming problem with fuzzy elements [31,32,35–37].

The task of fuzzy mathematical programming (FMP) is a task containing a target function or a vector of target functions (for example, local economic and environmental criteria) that need to be optimized and a system of inequalities or equalities describing conditions—constraints, part or all elements (criteria, restrictions, information about their importance, etc.) of the task are described indistinctly.

Consider the various situations that arise during the formalization of the waste management problem and for them we formulate the problem statement of the waste management tasks in the form of the FMP task and propose methods for solving them.

First we will pay attention to the situation when the task of fuzzy mathematical programming is posed for one criterion and several constraints, i.e., in a situation where the main criterion can be identified, to introduce the other criteria into the composition of the constraints, or it was possible to convolve the local criteria into one integrated criterion.

Suppose there is one normalized control criterion of the form $\mu_0(x)$ and several ($L$) fuzzy constraints. Before $f_q(x) \widetilde{>} b_q$, $q = \overline{1,L}$ we let, that the criterion is determined on the basis of the output indicators of the object, which are calculated using its mathematical models [38,39]. Suppose that the membership functions of the implementation of fuzzy constraints $\mu_q(x)$, $q = \overline{1,L}$ for each constraint are built in on the basis of expert evaluation with the involvement of specialists—experts, DM.

Let either the priority series be known $I = \{1, \ldots, L\}$, or the weight vector $\beta = \{\beta_1, \ldots, \beta_L\}$ for fuzzy constraints, reflecting the mutual importance of the constraints at the moment of setting the task.

Then, in general, the task of FMP for sustainable waste management:

$$\max_{x \in X} \mu_0(x)$$

$$\text{on } f_q(x) \widetilde{>} b_q, q = \overline{1,L},$$

can be written as:

$$\max_{x \in X} \mu_0(x),$$
$$X = \left\{ x : \operatorname*{argmax}_{x \in \Omega} \mu_q(x), q = \overline{1,L} \right\}$$

This formulation of the FMP problem with a clear objective function and fuzzy constraints with fuzzy instruction reflects the desire to maximize the objective function, fully satisfying the requirements of the constraints [40,41]. Assuming that all the membership functions are normal, i.e., their maximum values reaches unity, then the formulation of the FMP problem will take the form:

$$\max_{x \in X} \mu_0(x)$$
$$X = \left\{ x : x \in \Omega \wedge \mu_q(x) = 1, q = \overline{1,L} \right\}$$

We obtained a clear mathematical programming problem with maximization of the objective function on a $X$ clear set. Further assuming the concavity of the objective function $\mu_0(x)$, of constraints, $\mu_q$, $q = \overline{1,L}$ and the convexity of the admissible set $X$, then this problem can be solved by the ordinary methods of mathematical programming.

In practice, it is possible that the set $X$ is empty due to the lack of an alternative $x$ from the set X ($x \in X$) that simultaneously satisfies all constraints and, therefore, the problem has no solution.

In this case, it is necessary to abandon the clear solution of the original fuzzy problem and, taking advantage of the ambiguity of the constraints, set problems that take into account these fuzziness.

In this case, due to the impossibility of satisfying all criterion constraints, it is simultaneously necessary to use compromise schemes for taking into account the requirements of different criterion constraints. Let us use the ideas and schemes of trade-offs, incorporated in the direct methods multi-criteria evaluation of alternatives, to state the tasks of the FMP and to determine the solutions to these problems.

First, we reduce the initial problem to maximizing the objective function on the points of the Pareto set [42] formed by the constraints:

$$\max_{x \in X} \mu_0(x) \tag{1}$$

$$X = \left\{ x : \operatorname*{argmax}_{x \in \Omega} \sum_{q=1}^{L} \beta_q \mu_q(x) \wedge \sum_{q=1}^{L} \beta_q = 1 \wedge \beta_q \geq 0, \quad q = \overline{1,L} \right\} \tag{2}$$

The solution of this problem depends on the weight vector $\beta$ and consists of the vector of control actions (operational parameters, independent variables) $x^*(\beta)$, which provides the optimal values of the objective function $\mu_0(x^*(\beta))$ and the implementation of fuzzy constraints: $\mu_1(x^*(\beta)), \ldots, \mu_L(x^*(\beta))$.

Based on the application of the modified Pareto principle of optimality (PO), the following heuristic method of finding solutions for Equations (1) and (2) is proposed, based on the involvement of decision makers, experts, and consisting of the following main steps and points:

### 4.2. PO Method

(1) With the involvement of the decision maker to set $p_q$, $q = \overline{1, L}$—the number of steps for each $q$-th coordinate is given.

(2) Calculate $h_q = \frac{1}{p_q}$, $q = \overline{1, L}$—steps for changing the coordinates of the weight vector β.

(3) A set of weight vectors is constructed $\beta^1, \beta^2, \dots, \beta^N$, $N = (p_1 + 1) \cdot (p_2 + 1) \dots (p_L + 1)$ by varying the coordinates on the [0, 1] segments with the step $h_q$, defined in the previous paragraph.

(4) Based on the information received from the decision maker, specialists-experts determine the therm-set of fuzzy parameters and for each constraint membership functions the fuzzy constraints are constructed $\mu_q$, $q = \overline{1, L}$.

(5) N problems are solved when setting Equations (1) and (2) $\beta^t$, $t = \overline{1, N}$ and the current solution is determined: vector of mode, control parameters $x(\beta^t)$; criterion values $\mu_0(x(\beta^t))$ and membership functions of fuzzy constraints $\mu_1(x(\beta^t)), \dots, \mu_L(x(\beta^t))$.

(6) The resulting current decision is presented to the decision maker to select the final best solution. The best solution is chosen taking into account the preferences of the decision maker.

(7) If the current solution does not satisfy the decision maker, then they are assigned new values of the set of weight vectors, corrected $\beta^t$, $t = \overline{1, N}$ and returned to step 4. Otherwise, i.e., if the decision makers are satisfied with the current decisions, then go to step 8.

(8) The search for a solution is stopped, the results of the final selection are displayed, which satisfy the decision makers effective solutions: the optimal value of the regime (control) parameters—$x^*(\beta^t)$; providing the maximum value of the criterion—$\mu_0(x^*(\beta^t))$ and the maximum degree of perform fuzzy constraints—$\mu_1(x^*(\beta^t)), \dots, \mu_L(x^*(\beta^t))$.

In case of difficulties in the implementation of the last item, it is proposed to organize a dialogue procedure that allows the DM to receive additional information about its preferences, which substantially narrows the initial set of solutions.

In this method, the original Pareto set of solutions is approximated by $N$ points for which solutions are sought. The question of choosing the best solution in this and other proposed methods falls on the shoulders of the decision maker. There is a special method of interactive search for the best Pareto solution [43].

Assume that the priority series is known $I = \{1, \dots, L\}$. Now for the formulation of the problem of sustainable waste management we use the idea of the method of the main criterion.

For constraints, decision makers are assigned boundary values $\mu_q^R$, $q = \overline{1, L}$, that form constraints. The problem is solved in the following statement:

$$\max_{x \in X} \mu_0(x) \tag{3}$$

$$X = \left\{ x : x \in \Omega \wedge \arg\left(\mu_q(x) \geq \mu_q^R\right), \quad q = \overline{1, L} \right\}. \tag{4}$$

The solution to this problem depends on the boundary values: $x^*\left(\mu_1^R, \dots, \mu_L^R\right)$. This statement of the problem is more general than the original formulation of the FMP problem, and when $\mu_q^\Gamma = 1$, $q = \overline{1, L}$ transformed into Equations (1) and (2).

At the same time, it can be noted that arbitrariness is allowed in the designation of decision makers of boundary values $\mu_q^R$, $q = \overline{1, L}$. In order to reduce the arbitrariness and for greater validity, we recommend building interactive algorithms for assigning different boundary values, analyzing the decisions of decision makers, and selecting new boundary values. In the process of dialogue with the decision maker system, it studies the possibilities of obtaining different solutions, their sensitivity to

boundary values, and has the opportunity to influence the quality of decisions. Such opportunities are achieved through a dialogue with the decision maker, increasing its workload.

The modification of the presented problem statement is possible:

$$\max_{x \in X} \mu_0(x) \tag{5}$$

$$X = \left\{ x : x \in \Omega \wedge \arg\left(\mu_q(x) \geq \max_{x \in \Omega} \mu_q(x) - \Delta_q\right), \quad q = \overline{1, L} \right\}. \tag{6}$$

In this formulation, in contrast to the formulation of Equations (3) and (4), for each constraint, maximum values are determined, assignments are introduced $\Delta_q, q = \overline{1, L}$ (allowable deviations from maximum values), and the problem is solved on the resulting set of allowable values. At $\max_{x \in \Omega} \mu_q(x) = 1, \quad q = \overline{1, L}$, the statements of Equations (3) and (4) and Equations (5) and (6) coincide.

Let us give the structure of the proposed method for solving the problem of FMP for stable drilling waste control in Equations (3) and (4) and Equations (5) and (6).

*4.3. PO-Δ Method*

(1) Set a priority series for constraints $I = \{1, \ldots, L\}$.

(2) Based on the information received from the decision maker, specialist-experts to determine the term-set of fuzzy parameters and for each fuzzy restriction to build the membership functions of the implementation of restrictions $\mu_q(x), q = \overline{1, L}$.

(3) Decision makers assign initial boundary constraints $\mu_q^{R(r)}, q = \overline{1, L}, r = 1$ or determine maximum values $\max_{x \in \Omega} \mu_q(x), q = \overline{1, L}$ for each constraint and enter allowable deviations from maximum constraints (assignment) $\Delta_q, q = \overline{1, L}$ (for the case of setting (5) and (6)).

(4) Solve the problem of maximizing the objective function $\mu_0(x)$ taking into account the restrictions imposed in Equations (3) and (4) or (5) and (6), determine the current solutions:

$x\left(\mu_q^{R(r)}\right), \quad \mu_0\left(x\left(\mu_q^{R(r)}\right)\right), \quad \mu_1\left(x\left(x\left(\mu_1^{R(r)}\right), \ldots, \mu_L\left(x\left(\mu_q^{R(r)}\right)\right)\right), \quad q = \overline{1, L}$ (for staging (3)–(4)) or $x\left(\max\mu_q(x), \Delta_q\right), \mu_0\left(x\left(\max\mu_q(x), \Delta_q\right)\right), \mu_1\left(x(\max\mu_1(x), \Delta_1)\right), \ldots, \ldots, \mu_L\left(x(\max\mu_L(x), \Delta_L)\right), q = \overline{1, L}$ (when (5)−(6)).

(5) The resulting current solution is presented to the decision maker to select the final best solution satisfying it. If the current solution satisfies the decision maker, then the procedure for finding a solution is terminated and go to step 7.

(6) Otherwise, the decision maker is assigned the new values of the constraints $\mu_q^{R(r)}, r = r + 1$, and go to step 4.

(7) The final decisions are derived: the effective values of the control vector $x^*\left(\mu_q^R\right)$ or $\left(x^*\left(\max\mu_q(x), \Delta_q\right)\right)$ maximum values of the objective functions $\mu_0\left(x^*\left(\mu_q^{R(r)}\right)\right)$, or $\mu_0\left(x^*\left(\max\mu_q(x), \Delta_q\right)\right), q = \overline{1, L}$ maximum values of the membership function, describing the degree of implementation of fuzzy constraints $\mu_1\left(x^*\left(\mu_1^R\right)\right), \ldots, \mu_L\left(x^*\left(\mu_L^R\right)\right)$ or $\left(\mu_1(x^*(\max\mu_1(x), \Delta_1)), \ldots, \mu_L(x^*(\max\mu_L(x), \Delta_L))\right)$.

In practice, it is possible that the application of the Pareto principle of optimality is difficult or impossible, for example, with a large number of criteria and limitations. In this case, for the formulation and solution of the waste management problem, it is necessary to apply other principles of optimality. We present the problem statements with an approximate equality of the importance of fuzzy constraints in the form of the FMP problem based on the principle of equality:

$$\max_{x \in X_p} \mu_0(x) \tag{7}$$

$$X_p = \left\{ x : \underset{x \in X_1}{\operatorname{argmax}} \mu_1(x) \right\},$$
$$X_1 = \{ x : x \in \Omega \wedge \arg(\beta_1 \mu_1(x) = \ldots = \beta_L \mu_L(x)) \} \tag{8}$$

Changing the vector $\beta = (\beta_1, \ldots, \beta_L)$. The choice of the best solution can be made by the heuristic method based on the dialogue with the decision maker. The disadvantage of the task in Equations (7) and (8) is the predetermined nature of the obtained value $\mu_0(x)$ from the set $X$ (the set $X$ for a convex problem consists of one point).

The principle of quasi-equality allows, for a fixed β, to expand the set $X$:

$$\underset{x \in X}{\max} \mu_0(x) \tag{9}$$

$$X = \left\{ x : \arg\left( \|\mu(z) - \mu(x)\|_D \leq \delta_1 \right), z \in X_p \wedge \arg Y_\Pi \right\} \tag{10}$$

Here, the maximization $\mu_0(x)$ is carried out on the Pareto set in $\delta_1$—a neighborhood of the equality point $\mu(z)$ with coordinates $\mu_q(z)$, $q = \overline{1, L}$ that satisfy the condition $\beta_1 \mu_1(z) = \ldots = \beta_L \mu_L(z)$. The solution to Equations (9) and (10) depends from β and δ: $x^*(\beta, \delta)$. To choose a unique solution, we can construct an interactive algorithm.

The principle of quasi-equality can be expressed by varying the weight vector β:

$$\underset{x \in X}{\max} \mu_0(x) \tag{11}$$

$$X = \left\{ x : \underset{x \in X_1}{\operatorname{argmax}} \mu_1(x) \wedge \arg Y_\Pi \right\},$$
$$X_1 = \{ x : x \in \Omega \wedge \arg(\beta_1 \mu_1(x) = \ldots = \beta_L \mu_L(x)), \beta = (\beta_1, \ldots, \beta_L) \in B \},$$
$$B = \left\{ \beta : \|\beta - \beta^0\|_D \leq \delta_2 \right\} \tag{12}$$

where $\beta^0$ is the initial weight vector, $\delta_2$—determines the amount of allowable variation of the weight vector.

Based on the ideas of the equality principle (*E*) and the quasi-equality (*QE*), we propose the following method for solving Equations (7) and (8), (9) and (10), and (11) and (12).

### 4.4. E (QE) Method

(1) Based on the information received from the decision maker, specialists-experts determine the term-set of fuzzy parameters and for each fuzzy constraint construct the membership functions of the constraints implementation $\mu_q(x)$, $q = \overline{1, L}$.

(2) Set the values of the weight vector $\beta = (\beta_1, \ldots, \beta_L)$, providing $\beta_1 \mu_1(x) = .\beta_2 \mu_2(x) = \ldots = \beta_L \mu_L(x)$.

(3) Set the values of the allowable deviations $\delta_1$ and the allowable variation of the weight vector $\delta_2$ (for Equations (11) and (12)), select the type of metric $\|\mu(z) - \mu(x)\|_D \leq \delta_1$, $\|\beta - \beta^0\|_D \leq \delta_2$.

(4) Solve the problem of maximizing the objective function $\mu_0(x)$ taking into account the constraints imposed in Equations (7) and (8), (9) and (10), and (11) and (12), determine the current solutions: $x(\beta)$—values of the control (regime) parameters, $\mu_0(x(\beta))$ criterion values; $\mu_1(x(\beta)), \ldots, \mu_L(x(\beta))$—membership functions of perform fuzzy constraints (for Equations (7) and (8)); $x(\beta, \delta)$, $\mu_0(x(\beta, \delta))$, $\mu_1(x(\beta, \delta)), \ldots, \mu_L(x(\beta, \delta))$ (for Equations (9) and (10), (11) and (12)).

(5) Decisions to present the decision maker to select the final decision based on his preferences: $x^*(\beta)$; $\mu_0(x^*(\beta))$, $\mu_1(x^*(\beta)), \ldots, \mu_L(x^*(\beta))$ (in solving the problem in Equations (7) and (8)); $x^*(\beta, \delta)$, $\mu_0(x^*(\beta, \delta))$, $\mu_1(x^*(\beta, \delta)), \ldots, \mu_L(x^*(\beta, \delta))$ (in solving the problem in Equations (9) and (10)) or (11) and (12). If the current solutions do not satisfy the decision maker, they are assigned new values of β and (or) α and δ $(\delta_1, \delta_2)$ and the procedure for finding the best solution is repeated.

To present statements of the problem of sustainable management in the form of a fuzzy mathematical programming problem, which is most suitable for solving problems with strict environmental requirements?

- based on the principle of maximin (*MM*) (guaranteed result):

$$\max_{x \in X} \mu_0(x) \tag{13}$$

$$X = \left\{ x : \underset{x \in \Omega}{\arg\max} \min_q \beta_q \mu_q(x), q = \overline{1, L} \right\} \tag{14}$$

- based on the quasi-optimal maximin principle:

$$\max_{x \in X} \mu_0(x) \tag{15}$$

$$X = \left\{ x : \underset{x \in \Omega}{\arg\max} \min_q \beta_q \mu_q(x) - \Delta_q), q = \overline{1, L} \right\} \tag{16}$$

where $\Delta_q$—allowable deviations (concessions);
- based on the principle of sequential maximin:

$$\max_{x \in X_L} \mu_0(x) \tag{17}$$

$$
\begin{aligned}
&1. X_1 = \left\{ x : \underset{x \in \Omega}{\arg\max} \min_{q \in I} \beta_q \mu_q(x) \right\}; \\
&2. X_2 = \left\{ x : \underset{x \in X_1}{\arg\max} \min_{q \in I_1} \beta_q \mu_q(x) \right\}, \\
&\dots\dots \dots\dots \dots\dots \dots\dots \dots\dots \dots \\
&L. X_L = \left\{ x : \arg \max_{x \in X_{L-1}} \min_{q \in I_{L-1}} \beta_q \mu_q(x) \right\}
\end{aligned}
\tag{18}
$$

where *I*—full set of indexes (number restrictions), $I_1$—from the previous set *I* the number of the restriction that gave the solution to the previous maximin problem is excluded, $I_{L-1}$—The set consists of the last remaining restriction number;
- based on the quasi-optimal principle of sequential maximin:

$$\max_{x \in X_L} \mu_0(x) \tag{19}$$

$$
\begin{aligned}
&1. X_1 = \left\{ x : \underset{x \in \Omega}{\arg\max} \min_{q \in I} \beta_q \mu_q(x) - \Delta_1) \right\}; \\
&2. X_2 = \left\{ x : \underset{x \in X_1}{\arg\max} \min_{q \in I_1} \beta_q \mu_q(x) - \Delta_2) \right\}, \\
&\dots\dots \dots\dots \dots\dots \dots\dots \dots \\
&L. X_L = \left\{ x : \arg \max_{x \in X_{L-1}} \min_{q \in I_{L-1}} \beta_q \mu_q(x) - \Delta_L) \right\}
\end{aligned}
\tag{20}
$$

where $\Delta_q$—concessions, chosen by the decision maker in solving the *q*-th problem.
- based on the principle of absolute assignment (*AA*):

$$\max_{x \in X} \mu_0(x) \tag{21}$$

$$X = \left\{ x : x \in \Omega \wedge \arg\left( \sum_{q=1}^{L} \beta_q \mu_q(x) \geq \alpha \right), \alpha \in [0, 1] \right\} \tag{22}$$

- based on the principle of relative assignment (*RA*):

$$\max_{x \in X} \mu_0(x) \tag{23}$$

$$X = \left\{ x : x \in \Omega \wedge \arg\left( \prod_{q=1}^{L} \mu_q^{\beta q}(x) \geq \alpha \right), \alpha \in [0,1] \right\} \tag{24}$$

The last statement of the control problem in Equations (23) and (24) can be written, replacing the product with the sum of:

$$\max_{x \in X} \mu_0(x) \tag{25}$$

$$X = \left\{ x : x \in \Omega \wedge \arg\left( \sum_{q=1}^{L} \beta_q \log \mu_q(x) \geq \alpha \right), \alpha \in [0,1] \right\} \tag{26}$$

We present the main methods for solving problems in Equations (13) and (14), (15) and (16), (17) and (18), (19) and (20) (*MM* method), (21) and (22), and (23) and (24) (method *AA* (*RA*)-Δ).

### 4.5. MM Method

Clauses 1–4 are similar to the corresponding clauses of the PO methods.

(1) To set $p_q$, $q = \overline{1, L}$—number of steps for each $q$-th coordinate.

(2) Determine $h_q = \frac{1}{p_q}$, $q = \overline{1, L}$—steps for changing the coordinates of the weight vector β.

(3) A set of weight vectors is constructed: $\beta^1, \beta^2, \ldots, \beta^N$, $N = (p_1 + 1) \cdot (p_2 + 1) \ldots (p_L + 1)$ by varying the coordinates on the [0, 1] segments with a step $h_q$.

(4) Based on the information received from the decision maker, specialists-experts determine the therm-set of fuzzy parameters and for each constraint functions of the constraint fulfillment membership function are constructed $\mu_q$, $q = \overline{1, L}$.

(5) Solve the problem of maximizing the objective function: $\max \mu_0(x)$ on the set, which is determined depending on the chosen principle: maximin in Equations (13) and (14), the quasi-optimal principle of maximin in Equations (15) and (16), sequential maximin in Equations (17) and (18) and quasi-optimal sequential maximin in Equations (19) and (20) determine the solutions: $x(\beta)$—control vector values; $\mu_0(x(\beta))$—criteria and $\mu_1(x(\beta)), \ldots, \mu_L(x(\beta))$—the membership function (degree) of the fulfillment of fuzzy restrictions in Equations (13) and (14), (17) and (18) or $x(\beta, \Delta_q)$, $\mu_0(x(\beta, \Delta_q))$, $\mu_1(x(\beta, \Delta_q)), \ldots, \mu_L(x(\beta, \Delta_q))$ (at the settings in Equations (15) and (16), (19) and (20)).

(6) Current decisions to present the decision maker to choose the final decision based on his preferences. If the current decisions satisfy the decision maker, then go to step 8.

(7) If the current solutions do not satisfy the decision maker, they are assigned new values of β and (or) assignments $\Delta_q$ and return to step 5 in order to conduct a new procedure for finding the best solution.

(8) The search for a solution to stop; The conclusion of the final decision of the chosen decision makers $x^*(\beta)$—optimal values of the control vector; $\mu_0(x^*(\beta))$—maximum criterion value; $\mu_1(x^*(\beta)), \ldots, \mu_L(x^*(\beta))$—membership functions of performing fuzzy constraints (when solving problems (13) and (14), (17) and (18)); or similarly $x^*(\beta, \Delta_q)$, $\mu_0(x^*(\beta, \Delta_q))$, $\mu_1(x^*(\beta, \Delta_q)), \ldots, \mu_L(x^*(\beta, \Delta_q))$ (in solving the problem in Equations (15) and (16) or (19) and (20)).

### 4.6. Method AA(RA)-Δ

Clauses 1–4 coincide with the corresponding clauses of the *PO, MM* methods.

(1) Sets $p_q$, $q = \overline{1, L}$—the number of steps for each $q$-th coordinate.

(2) It is determined $h_q = \frac{1}{p_q}$, $q = \overline{1,L}$—the steps for changing the coordinates of the weight vector β.

(3) A set of weight vectors is constructed $\beta^1, \beta^2, \ldots, \beta^N$, $N = (p_1 + 1) \cdot (p_2 + 1) \ldots (p_L + 1)$ by varying the coordinates on the [0, 1] segments with a step $h_q$.

(4) Based on the information received from the decision maker, specialists-experts determine the therm-set of fuzzy parameters and for each constraint, the membership functions of the constraints are built.

(5) Solve $N$ problems $\max_{x \in X} \mu_o(x)$ on the set $X$ defined by Equations (22) or (24). Identify current solutions: values of control parameters; $x(\beta)$—values of control parameters; $\mu_0(x(\beta))$—criterion value and $\mu_1(x(\beta)), \ldots, \mu_L(x(\beta))$—degree of fulfillment of fuzzy constraints.

(6) If the solution obtained in the previous clauses satisfies the decision maker to stop the search for a solution and display the results go to clauses 8, otherwise proceed to clauses 7.

(7) In order to improve solutions the decision maker corrects the values of β and return to clauses 5.

(8) Stop the search for the solution and the conclusion of the final decisions: $x^*(\beta)$ (the optimal values of the independent variables—control, operating parameters), providing the extreme value of the criterion $\mu_0(x^*(\beta))$ and $\mu_1(x^*(\beta)), \ldots, \mu_L(x^*(\beta))$ (the degree of implementation of fuzzy constraints).

Thus, based on the application of various principles of optimality, more suitable for different situations or by combining them can obtain various formulations of the problem of sustainable waste management in a fuzzy environment and develop methods for solving them.

## 5. Conclusions

The paper proposes a method for modeling and sustainably managing the process of disposal of drill cuttings in marine conditions in a fuzzy environment using a fuzzy inference system. The proposed approach allows us to overcome the problems of shortage and fuzziness of the initial information due to the knowledge and experience of specialists-experts of subject area and DM. The main procedures of the method are based on the Mamdani algorithm and are implemented in the MatLab environment using Fuzzy Logic Toolbox. Based on the results obtained, it can be concluded that under conditions of fuzzy information about the input and output parameters of the system, the developed fuzzy rules describe the dependence between the input and output fuzzy parameters quite well.

Based on an expert assessment, real fuzzy values of input (volume of chemical reagents and injection pressure) and output (disposal of drill cuttings) parameters that describe fuzzy dependence between input and output fuzzy parameters were selected. An expert assessment was also organized and conducted to determine five levels of fuzzy parameter values, for which membership functions were built using the Fuzzy Logic Toolbox application. Based on the production model of knowledge representation, a set of rules for a fuzzy inference system has been developed. Thus, linguistic models are constructed that describe the dependence between the real values of the input and output linguistic variables. As an advantage of the results obtained, it can be noted that the developed rule base allows adequate modeling of the process of drilling waste disposal. This means practical importance and confirms the advantages of the proposed fuzzy approach for managing drill cuttings in marine conditions. In carrying out further research, it is planned, in the conditions of the availability of reliable source quantitative data, to solve the modeling problem using a deterministic approach. And with the fuzziness of the initial data, it is planned to build other forms of the membership function of the fuzzy parameters and carry out fuzzy modeling based on the rule base.

The purpose of the study is to analyze the problems of sustainable management of drill cuttings during drilling in marine conditions and to develop an approach to their solution using a fuzzy approach. The contribution of the expected results of the study is to solve the problems of waste management in marine conditions (for example, marine fields of the Caspian Sea) and to increase the adequacy of the sustainable management process by using the experience and knowledge of specialist-experts (fuzzy information) and the apparatus of the theory of fuzzy sets.

Mathematical formulation of fuzzy programming problems for sustainable waste management in a fuzzy environment has been formulated and obtained and heuristic methods for solving them have been developed. The obtained statements of the problem of sustainable waste management in various situations arising from waste management according to environmental and economic criteria and the available information necessary to solve the problem are reduced to the tasks of fuzzy mathematical programming while preserving their vagueness. Based on the modification of various trade-off schemes: Pareto optimality; equality and quasi-equality; maximin; absolute and relative assignment to work in a fuzzy environment, developed methods for their solutions which are presented in the form of heuristic methods. The originality and scientific novelty of the results lies in the fact that the problem is posed and solved in a fuzzy environment without prior conversion to deterministic problems. This provides a more complete use of the collected fuzzy information and obtaining an adequate solution to the problem of production waste management when the initial information is unclear, that is, in real situations.

As the main limitation of the implementation of the proposed fuzzy approaches in the modeling and management of drilling waste management processes, can note the presence and accessibility of specialist-experts subject area. But as practice shows, such specialist-experts are available. And the problems of obtaining knowledge from them are solved using the correct application of expert assessment methods and theories of fuzzy sets. In the future, the authors plan to use the proposed decision-making methods for managing various industrial wastes.

A feature of the work is the practical applicability of the proposed methods of fuzzy modeling of drill cuttings control processes in marine conditions, as well as methods for setting and solving decision problems in a fuzzy environment by reducing them to fuzzy mathematical programming problems.

**Author Contributions:** Conceptualization, B.O. and S.S.; formal analysis, A.Z.; resources, K.D. and X.-G.Y.; writing—original draft preparation, B.O., A.Z., S.S. and X.-G.Y.; writing—review and editing, R.K.A. and J.F.

**Funding:** This research was funded by the Construction Project of Modern Seed Industry Innovation System for Characteristic Crops in Qinghai Plateau (Project No. 2017-NK-A7). And the APC was funded by L.N.Gumilyov Eurasian National University.

**Acknowledgments:** We deeply thank anonymous reviewers for their insightful suggestions and constructive comments, and we are also grateful to the editors for their editing to our manuscript.

**Conflicts of Interest:** The authors declare no conflict of interest.

## Abbreviations

KSCS—Kazakhstan sector of the Caspian Sea
FST—Fuzzy set theory
EDM—Experts decision maker
E. Mamdani—Ebrahim Mamdani

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
