# Peer review of "Sustainable Waste Management Drilling Process in Fuzzy Environment"

_sustainability, doi:10.3390/su11246995_

Round 1

Reviewer 1 Report

The proposed paper addresses a technical and specific field.

It is suggested to define clearly the research question, explaining the contribution in relation to the existing literature.

Author Response

Dear Reviewer,

Thank you.

The aim of the study is to analyze the problems of sustainable management of drill cuttings during drilling in marine conditions based on literature data and to develop an approach to their solution using a fuzzy approach.

The contribution of the expected results of the study is to solve the problems of waste management in marine conditions (for example, offshore fields of the Caspian Sea) and to increase the adequacy of the sustainable management process by using the experience and knowledge of specialists-experts (fuzzy information) and the apparatus of the theory of fuzzy sets.

Thus, the research question is more clearly defined in the article, the contribution of the work as applied to existing literature is disclosed.

All the very best,

Authors.

Reviewer 2 Report

The topic of the paper is modeling and control of the drill cutting process disposal in a fuzzy environment. The paper is divided into two main subjects. One is the fuzzy inference system with the linguistic variables associated with the assessment of drill cuttings disposal. The inference system based on the Mamdani model is well known and used in the literature for years. In this aspect, this part of paper has no unique contribution, except a very simple application approach.
In addition, section 3 has major shortcomings.
Tables 1-2 show 5 linguistic values for the variables, while in Figures 3-4 there are only 3.
Table 2 shows universums for fuzzy variables, while the parameters of the membership functions are more important (this is missing).
The shapes of membership functions raise interpretation doubts. Please explain the desirability of not overlapping membership functions for input variables and the lack of external left and right functions for output variables.
The second part of paper is devoted to fuzzy mathematical programming. This part, in turn, is a purely theoretical approach, without reference to the topic of sustainable waste management in the drilling process.
Both parts of the article are not consistent, for example there is no consistency in the symbols (beta once means fuzzy parameters, once weight vector). The paper is not clearly presented, its organization and technical content is very poor. Not all symbols used are described. The formulas are no numbered.
The Authors should make a clear discussion versus the unique contribution of the paper (in the introduction), especially in comparison to articles about the same subject.
The conclusion section could be rewritten in order to: discuss research contributions and indicate practical advantages, discuss research limitations and supply several solid and insightful future research suggestions.
The Authors should to revise all document and some mistakes should be corrected (superfluous hyphens [line 26,36,39,41,44,45 …], superfluous number [line 288], superfluous colon [line 291], unknown symbol [line 427], double dots [line 430] …).

Author Response

Dear Reviewer,

Thank you. 

The topic of the paper is modeling and control of the drill cutting process disposal in a fuzzy environment. The paper is divided into two main subjects. One is the fuzzy inference system with the linguistic variables associated with the assessment of drill cuttings disposal. The inference system based on the Mamdani model is well known and used in the literature for years. In this aspect, this part of paper has no unique contribution, except a very simple application approach.

In addition, section 3 has major shortcomings. Tables 1-2 show 5 linguistic values for the variables, while in Figures 3-4 there are only 3.

Table 2 shows universums for fuzzy variables, while the parameters of the membership functions are more important (this is missing). The shapes of membership functions raise interpretation doubts. Please explain the desirability of not overlapping membership functions for input variables and the lack of external left and right functions for output variables.
The second part of paper is devoted to fuzzy mathematical programming. This part, in turn, is a purely theoretical approach, without reference to the topic of sustainable waste management in the drilling process.

Both parts of the article are not consistent, for example there is no consistency in the symbols (beta once means fuzzy parameters, once weight vector). The paper is not clearly presented, its organization and technical content is very poor. Not all symbols used are described. The formulas are no numbered.

The Authors should make a clear discussion versus the unique contribution of the paper (in the introduction), especially in comparison to articles about the same subject.
The conclusion section could be rewritten in order to: discuss research contributions and indicate practical advantages, discuss research limitations and supply several solid and insightful future research suggestions.

The Authors should to revise all document and some mistakes should be corrected (superfluous hyphens [line 26,36,39,41,44,45 …], superfluous number [line 288], superfluous colon [line 291], unknown symbol [line 427], double dots [line 430] …).

Answer: The contribution of this part of the article really consists in applying a fuzzy approach to solving the problems of managing drill cuttings in a particular marine field of the Northern Caspian (Kashagan field) based on the adaptation of the Mamdani algorithm.

In addition, section 3 has major shortcomings. Tables 1-2 show 5 linguistic values for the variables, while in Figures 3-4 there are only 3.

Answer: Indeed, Tables 1 and 2 show the 5th level of values of fuzzy parameters (HG-High; НМ- High of medium; MD- Medium: LM-Low of medium: LW-Low), i.e. values of linguistic variables, and Figure 4 shows 3 linguistic parameters  (volume of chemical reagents),  (injection pressure) and  (disposal of drill cuttings). Figures 5 a, b, c are added with additional linguistic meanings and these figures are replaced in the paper.

Table 2 shows universums for fuzzy variables, while the parameters of the membership functions are more important (this is missing).

Answer: Table 2 shows universes for fuzzy variables, and the parameters of the membership function, i.e. the main points describing membership functions are derived from expert judgment. Then, smoothly connecting these points, the graphs of the membership functions are obtained, which are identical to the graphs of the membership function shown in figures 5 a, b, c.

The shapes of membership functions raise interpretation doubts. Please explain the desirability of not overlapping membership functions for input variables and the lack of external left and right functions for output variables.

Answer: The forms of the membership functions on the figures 5 a, b, c. are changed, brought into line with the data of table 2. Non-overlapping membership functions for input variables are obtained on the basis of expert data taking into account the universals given in table 2, and the membership function of the output variable is corrected and the correct form is shown in figure 5 c.

The second part of paper is devoted to fuzzy mathematical programming. This part, in turn, is a purely theoretical approach, without reference to the topic of sustainable waste management in the drilling process.

Answer: In the second part of the paper, an approach to the development of theoretical basis of waste management problems in the form of fuzzy mathematical programming problems and applying them to solve the problems of decision-making on the management of drill cuttings is proposed. That is, the formulation of waste management problems under different conditions in the form of fuzzy mathematical programming problems is obtained, and heuristic methods for their solution are proposed. Currently, the authors are working on the application of the proposed approach to solve a specific problem in the conditions of the Kazakhstan sector of the Caspian sea.

Both parts of the article are not consistent, for example there is no consistency in the symbols (beta once means fuzzy parameters, once weight vector).

Answer: To solve the problems of inconsistency in the characters fuzzy parameters (the first part of the paper) are re-marked with other symbols and and the weight vector in the second part is indicated by  (beta).

The paper is not clearly presented, its organization and technical content is very poor. Not all symbols used are described. The formulas are no numbered.

Answer: Work has been carried out to clearly present the article, to improve its organization and technical content. All symbols used are described, all formulas referenced are numbered.

The Authors should make a clear discussion versus the unique contribution of the paper (in the introduction), especially in comparison to articles about the same subject.

Answer: In the introduction of the paper the information on discussion and comparison of the contribution of the paper with the articles on the topic of research is added, the literature is added and references to new sources are made.

The conclusion section could be rewritten in order to: discuss research contributions and indicate practical advantages, discuss research limitations and supply several solid and insightful future research suggestions.

Answer: Section Conclusion(s) modified, added info for the discussion of the contribution to research, highlighted the practical importance and the benefits of working, discuss the limitations in research and offer the necessary suggestions for future research on the topic of the article.

The Authors should to revise all document and some mistakes should be corrected (superfluous hyphens [line 26,36,39,41,44,45 …], superfluous number [line 288], superfluous colon [line 291], unknown symbol [line 427], double dots [line 430] …).

Answer: Fixed mistakes (the extra hyphens [on line 26,36,39,41,44,45,47,52,55,58,60, 62, 66, 67, 76, 81, 88, 96, 98, 102-105, 108, 114, 121, 137, 151,166, 168, 171], the number 6 on line 288 means Figure 6,  the number of the picture on the previous line did not fit], extra colon on line 291 deleted, unknown character in line 389, 414, 419, 426, 428, 430 «Г – superscript», fixed on R, indicates the boundary value], double points in the line 428 remote...).

All the very best,

Authors

Reviewer 3 Report

This research is an interesting topic related to the sustainable management of waste during drilling in marine conditions. The paper is structurally complete and the research results are specific.
1. In the introduction chapter, the content of the literature should be strengthened to highlight the gaps in the literature.
2. In the second chapter, the content of the fuzzy environment should be strengthened to highlight the suitability of the fuzzy method in Chapter 3.
3.Please check the parameter description of the equation in the full text, if there are any omissions or explanations.
4.The diagrams, tables and equations in Chapters 3 and 4 should be elaborated. Some of the contents are not clear.
5.The conclusion should include following contents such as background, research objective, experiment result, finding and future research and limitations. The practical applicability of the paper is an implicit aim of the paper, and should be better highlighted and evaluated, especially in the final part.

Author Response

Dear Reviewer,

Thank you.

This research is an interesting topic related to the sustainable management of waste during drilling in marine conditions. The paper is structurally complete and the research results are specific.

In the introduction chapter, the content of the literature should be strengthened to highlight the gaps in the literature. In the second chapter, the content of the fuzzy environment should be strengthened to highlight the suitability of the fuzzy method in Chapter 3.

3.Please check the parameter description of the equation in the full text, if there are any omissions or explanations.

4.The diagrams, tables and equations in Chapters 3 and 4 should be elaborated. Some of the contents are not clear.

5.The conclusion should include following contents such as background, research objective, experiment result, finding and future research and limitations. The practical applicability of the paper is an implicit aim of the paper, and should be better highlighted and evaluated, especially in the final part.

In the introduction chapter, the content of the literature should be strengthened to highlight the gaps in the literature.

Answer 1:  In the introduction to emphasize the problems of the study, the content of the literature is strengthened, references are made to additional literature [6, 7, 10, 13].

In well-known formulations and solution methods fuzzy problems at the formulation stage are replaced by a system of equivalent clear problems based on a set of level α. This approach allows to get an approximate solution to the fuzzy problem, but a significant part of the original collected fuzzy information is not used and is lost, which will lead to a decrease in the adequacy of the obtained solutions. The advantage of the fuzzy approach proposed in this paper to the formulation and solution of fuzzy mathematical programming problems is that the problem is posed and solved in a fuzzy environment without first converting to equivalent clear problems.

 This allows you to save and maximize the use of the original fuzzy information. And this, in turn, allows you to get an adequate solution to the problem of managing production waste in a fuzzy environment, i.e. in real situations.

In the second chapter, the content of the fuzzy environment should be strengthened to highlight the suitability of the fuzzy method in Chapter 3.

Answer 2: In the second section, the content of the fuzzy environment, sources and reasons for the fuzziness of some part of the initial information are strengthened, i.e. the suitability and necessity of the fuzzy method in the section 3 is substantiated

To study the method of re-injection of drill cuttings and its technology, it is required to develop a system of mathematical models that describe the processes of pumping drill cuttings and control these processes. As shown by the practice of collecting and processing the necessary information for the development of mathematical models of drill cuttings management processes in offshore oil and gas fields of KSCS, there is a shortage of reliable theoretical and statistical information necessary for constructing analytical or statistical models. The organization and conduct of theoretical and experimental studies of these processes in order to collect the missing part of the information is not economically feasible. But during the analysis, it turned out that there are experienced specialists in the subject area, i.e. drillers, ecologists and engineers who can adequately describe in a natural language the processes of pumping drill cuttings and controlling these processes. They can verbally describe what are the main parameters and how they affect the process of disposal of drill cuttings. Based on such fuzzy information using the theory of fuzzy sets, one can construct rule bases, i.e. linguistic models of the drilling waste disposal process, which are relatively easy to implement using the MatLab Fuzzy Logic Toolbox application.

Please check the parameter description of the equation in the full text, if there are any omissions or explanations.

Answer 3: The parameters of the equation in the text are checked, some parameters and designations are described.

The diagrams, tables and equations in Chapters 3 and 4 should be elaborated. Some of the contents are not clear.

Answer 4: Some figures and equations given in sections 3 and 4 are further explained.

As can be seen from Figure 3, the input parameters for the control object i.e. for the installation of injection of drill cuttings in marine field values of the injection pressure and volume of chemical reagents, and the output parameter of the installation is an indicator of the utilization of drill cuttings.

Since these parameters are difficult to quantify quantitatively, it is advisable to evaluate their values using expert evaluation (in the measurement, evaluation block). Then, the fuzzy output system receives fuzzy values of the estimated input parameters, and the fuzzy value of the percentage of utilization is obtained from the output of the system. Depending on the output value of the fuzzy inference system and the rule base, control actions are generated.

The conclusion should include following contents such as background, research objective, experiment result, finding and future research and limitations. The practical applicability of the paper is an implicit aim of the paper, and should be better highlighted and evaluated, especially in the final part.

Answer 5:  In conclusion, the level of achievement of the research goal is estimated, the main results obtained are summarized, the studies planned in the future on the application of the proposed decision-making methods for waste management, as well as possible limitations in the implementation of the proposed methods are highlighted. The practical applicability of the research results in waste management in a fuzzy environment is substantiated.

The aim of the study was to analyze the problems of sustainable management of drill cuttings during drilling in marine conditions and to develop an approach to their solution using a fuzzy approach.

In the future, the authors plan to use the proposed decision-making methods for managing various industrial wastes.

As the main limitation of the implementation of the proposed fuzzy approaches when modeling and managing drilling waste management processes, the presence and accessibility of subject matter experts- specialists are noted.

A feature of the work is the practical applicability of the proposed methods of fuzzy modeling of drill cuttings control processes in marine conditions, as well as methods for setting and solving decision-making problems in a fuzzy environment by reducing them to fuzzy mathematical programming problems.

All the very best,

Authors

Round 2

Reviewer 2 Report

Thank you for the answers. I am satisfied with most of the corrections made.

Unfortunately, I am still not sure if the fuzzy approach for system inputs and outputs is well understood here. You wrote "Non-overlapping membership functions for input variables are obtained on the basis of expert data taking into account the universals". In this context, you can apply a deterministic knowledge base (expert systems) and appropriate inference (without using fuzzy sets).
Another issue, the overlapping in membership functions of fuzzy sets makes the output surface smooth so that jerky system responses are avoided.

Author Response

Reply to the comments and recommendations of the reviewer.

Review

”Unfortunately, I am still not sure if the fuzzy approach for system inputs and outputs is well understood here. You wrote "Non-overlapping membership functions for input variables are obtained on the basis of expert data taking into account the universals". In this context, you can apply a deterministic knowledge base (expert systems) and appropriate inference (without using fuzzy sets).

Another issue, the overlapping in membership functions of fuzzy sets makes the output surface smooth so that jerky system responses are avoided.”

Answer: In fact, the experts evaluated the main control points of the membership function, specifically: the extreme points where the membership functions take minimum values, the point in the middle of the universe where the membership functions take maximum values, and also the points between the maximum and minimum values of the membership function. Then, by smoothly connecting these points, graphic forms of triangular form membership functionы are obtained, which are easily approximated to analytical expressions. Moreover, in the MatLab Fuzzy Logic Toolbox application such forms of membership function are available and it is convenient to apply them. In our case, these forms of membership functions, and the results of fuzzy modeling based on the constructed rule base, gave fairly adequate (similar) data with real data. This allows using the Fuzzy Logic Toolbox application to carry out fuzzy simulations with other fuzzy values of the input linguistic variables. Therefore, in this paper, a fuzzy approach is used. In the paper, these moments are described additionally. You are right that a deterministic base could be applied without using fuzzy sets. Thank you very much, we plan to investigate such an approach in further studies.

Also, in further studies, we plan, at your advice, to choose linear and nonlinear overlapping membership functions to ensure surface smoothness and avoid sharp system responses. Thanks for the comments and for the valuable advice.

Certain work has been done to improve the design of the study. In order to confirm the conclusions with the results, the section “Conclusion” provides additional information and areas of further research are identified.

Authors.